# Efficacy of Removing Thermafil and GuttaCore from Straight Root Canal Systems Using a Novel Non-Surgical Root Canal Re-Treatment System: A Micro-Computed Tomography Analysis

**DOI:** 10.3390/jcm10061266

**Published:** 2021-03-18

**Authors:** Vicente Faus-Llácer, Rubén Linero Pérez, Ignacio Faus-Matoses, Celia Ruiz-Sánchez, Álvaro Zubizarreta-Macho, Salvatore Sauro, Vicente Faus-Matoses

**Affiliations:** 1Department of Stomatology, Faculty of Medicine and Dentistry, University of Valencia, 46010 Valencia, Spain; fausvj@uv.es (V.F.-L.); rubenlinero@hotmail.com (R.L.P.); ignacio.faus@uv.es (I.F.-M.); ceruizsan@gmail.com (C.R.-S.); vicente.faus@uv.es (V.F.-M.); 2Department of Endodontics, Faculty of Health Sciences, Alfonso X El Sabio University, 28691 Madrid, Spain; 3Department of Surgery, Faculty of Medicine and Dentistry, University of Salamanca, 37008 Salamanca, Spain; 4Dental Biomaterials and Minimally Invasive Dentistry, Department of Dentistry, Cardenal Herrera-CEU University, 46115 Valencia, Spain; salvatore.sauro@uchceu.es; 5Department of Therapeutic Dentistry, I.M. Sechenov First Moscow State Medical University, 119146 Moscow, Russia

**Keywords:** endodontics, GuttaCore, micro-computed tomography, non-surgical root canal re-treatment, root canal filling material, Thermafil

## Abstract

The present study aims to evaluate the effectiveness of an XP-endo non-surgical root canal re-treatment system in removing both GuttaCore and Thermafil gutta-percha carrier-based root canal filling materials from straight root canal systems using micro-computed tomography (micro-CT) analysis. The study was performed on 20 single-rooted upper teeth, which were randomly allocated into the following study groups: Group A, Thermafil and AH Plus sealer (*n* = 10); Group B, GuttaCore and AH Plus sealer (*n* = 10). Before and after the non-surgical root canal re-treatment procedure, the samples were submitted for a micro-CT analysis. The volume of the root canal filling material (mm^3^), the volume of the remaining root canal filling material (mm^3^) and the time (minutes) needed to remove the root canal filling material were also recorded. Student’s *t*-test was used to analyze the results. No statistically significant differences were found between the volume of the remaining root canal filling material in the GuttaCore and Thermafil root canal filling systems at the coronal third (*p* = 0.782), middle third (*p* = 0.838) or apical third (*p* = 0.882) of the straight root canal systems; however, the GuttaCore required a statistically significant (*p* = 0.037) shorter amount of time (4.72 ± 0.76 min) to be removed than the Thermafil carrier-based root canal filling material (5.92 ± 1.42 min). The XP-endo Finisher non-surgical endodontic re-treatment system removes both GuttaCore and Thermafil gutta-percha carrier-based root canal filling materials from straight root canal systems, although removal of the GuttaCore gutta-percha carrier-based root canal filling material required less time.

## 1. Introduction

At present, root canal treatment has a reported survival rate of up to 84.1% [1] to 97.3% [2]; however, intraoperative complications during the endodontic therapy can potentially lead to treatment failure [3], continued primary endodontic infection [4,5] and the possible need for subsequent treatment options including non-surgical root canal re-treatment, endodontic surgery, autotransplantation, intentional replantation or tooth extraction [6,7,8,9]. Root canal treatment failure is frequently linked to insufficient disinfection of the root canal system, the root canal filling material overextending through the apical foramen, inadequate obturation, unsuitable coronal restoration, untreated canals and iatrogenic procedural errors due to natural cavity design with limited access or instrumentation complications (ledges, perforations or fractured instruments) [3,10,11,12,13]. Therefore, non-surgical root canal re-treatment is recommended as a first treatment approach for the removal of root canal filling material, enabling thorough cleaning, disinfection and re-instrumentation of the root canal system [14,15,16]. Numerous non-surgical root canal re-treatment techniques and materials have been proposed, including stainless steel hand files, nickel titanium (NiTi) endodontic rotary and reciprocating instruments, specialized NiTi endodontic re-treatment files [17], ultrasonic appliances, Gates Glidden burs or a combination thereof. In addition, the adjuvant use of gutta-percha solvent agents or heat application has also been proposed for facilitating access to the root canal system for removal of the root canal filling material [18] in order to enable proper cleaning, shaping and disinfection of the root canal system. Although many non-surgical root canal re-treatment techniques can be used to remove root canal filling material within the root canal system, none of them are capable of completely removing all the aforementioned filling material [17]; therefore, novel non-surgical root canal re-treatment systems have been proposed, such as the XP-Endo Retreatment System (FKG Dentaire, La-Chaux-de-Fonds, Switzerland), which is composed of three endodontic rotary files: the DR1 endodontic rotary file has been designed with alternating cutting edges and a triangular cross-section and the XP-endo Shaper and XP-endo Finisher R endodontic rotary files are made of NiTi MaxWire alloy (Martensite-Austenite Electropolish Flex), which increases the efficacy of root canal filling removal due to its shape memory and austenite phase conversion [19].

Many root canal filling materials and techniques for sealing the root canal system have been described; however, lateral condensation remains the most commonly used technique for root canal filling [20,21]. The lack of homogeneity in the root canal system has led to the development of new root canal filling techniques, including warm gutta-percha techniques such as gutta-percha carrier-based root canal filling techniques and vertical condensation root canal filling techniques [22,23]. Gutta-percha carrier-based root canal filling materials have become widely used [24,25] due to easy handling [26] and high marginal sealing capacity [27]; however, the rigid carrier of these root canal filling techniques’ results is difficult to completely remove, especially the Thermafil carrier-based root canal filling system. Therefore, a novel carrier-based root canal filling system has been developed, with a special carrier based on a cross-linked gutta-percha core; this should be easier to remove compared to the Thermafil carrier-based root canal filling system [28].

The present study aims to analyze the efficacy of the XP-endo non-surgical root canal re-treatment system in removing two types of gutta-percha carrier-based root canal filling materials, Thermafil and GuttaCore, from straight root canal systems using a micro-computed tomography (micro-CT) analysis. The null hypothesis (H_0_) holds that there is no difference between the gutta-percha carrier-based root canal filling materials during removal from straight root canal systems.

## 2. Materials and Methods

### 2.1. Study Design

Twenty (20) single-rooted anterior teeth (upper canine) were studied. The teeth were extracted due to periodontal concerns. The studied teeth had curvatures of <10°, measured using Schneider’s method [29], mature roots and no prior root canal filling materials, root resorptions or calcium metamorphosis. The sample teeth were chosen from the Department of Stomatology at the University of Valencia (Valencia, Spain) during the period from September to October 2020. An ANOVA one-way test was used to calculate the samples. A randomized controlled experimental trial was carried out following the principles outlined in the statement by the German Ethics Committee (Zentrale Ethikkommission, 2003) on using organic tissues for the purposes of medical research. The Ethics Committee of the University of Valencia approved the study (Process no. 19/2020). All patients provided informed consent for their teeth to be used in the study.

### 2.2. Experimental Procedure

Digital radiographs were taken of the teeth in the buccolingual and mesiodistal directions so as to ensure that the root axes were straight and the root canal sections were slightly oval. The teeth were all decoronated using a diamond disc (Brasseler, GA, USA, Savannah, GA, USA) to obtain a standardized root length of 16 mm. To ascertain the apical patency, a 10-K file (Dentsply Maillefer, Baillagues, Switzerland) was used until the file could be seen at the apical foramen when magnified (OPMI pico, Zeiss Dental Microscopes, Oberkochen, Germany). A 15-K file was used to confirm the glide path (Dentsply Maillefer, Baillagues, Switzerland). Afterwards, root canal procedures were carried out using the ProTaper Gold endodontic rotary system (Dentsply Maillefer, Baillagues, Switzerland) set to a 25.08 F2 file with a 6:1 reduction handpiece (X-Smart plus, Dentsply Maillefer, Baillagues, Switzerland). The torque-controlled motor was set to continuously rotate at 300 rpm and 2 N/cm torque, following the recommendations provided by the manufacturer. The root canal systems were then irrigated using 5 mL of a solution with 5.25% sodium hypochlorite (NaOCl; Clorox, Oakland, CA, USA). The final irrigation consisted in 5 mL of 5.25% NaOCL (Clorox, Oakland, CA, USA), 5 mL of 17% ethylenediaminetetraacetic acid (EDTA) (SmearClear, SybronEndo, CA, USA), 5 mL of 5.25% NaOCl (Clorox, Oakland, CA, USA) and 5 mL of sterile saline solution (Braun^®^, Melsungen, Germany). A specialized endodontic needle (Miraject Endo Luer, Hager and Werken, Duisburg, Germany) was inserted up to 1 mm of the working length. The root canal systems were then dried using sterile paper points (Dentsply Maillefer, Ballaigues, Switzerland). Next, the root canal systems were randomly sealed (Epidat 4.1, Galicia, Spain) into one of the following warm gutta-percha carrier-based root canal filling systems: Group A, Thermafil (Dentsply Maillefer, Ballaigues, Switzerland) along with an epoxy-amine resin-based sealer (AH Plus, Dentsply DeTrey, Konstanz, Germany) (*n* = 10); or Group B, GuttaCore (Dentsply Maillefer, Ballaigues, Switzerland) along with an epoxy-amine resin-based sealer (AH Plus, Dentsply DeTrey, Konstanz, Germany) (*n* = 10). This was achieved by heating a GuttaCore obturator (size 25) in a ThermaPrep heater obturator oven (Dentsply Maillefer, Ballaigues, Switzerland) before inserting it in the root canal system up to the working length. A periapical radiograph of the buccolingual projection was taken to assess the quality of the root canal filling; if any voids were detected, these specimens were discarded and replaced. Lastly, Cavit^TM^ (3M ESPE, Saint Paul, MN, USA) was used to temporarily seal the access cavity, and the specimens were stored in a 37 °C atmosphere with 100% humidity for 14 days.

The gutta-percha carrier-based root canal filling materials were removed, while the time needed was registered with a chronometer, under magnification (OPMI Pico, Zeiss Dental Microscope, Oberkochen, Germany) using the XP-Endo Retreatment System (FKG Dentaire, La-Chaux-de-Fonds, Switzerland), starting with continuous rotation with the DR1 endodontic rotary file (FKG Dentaire, La-Chaux-de-Fonds, Switzerland) at 800 rpm and 1.5 N/cm torque inserted in the gutta-percha carrier-based root canal filling materials so as to create a 3–4-mm space from the cementoenamel junction. Afterwards, an XP-endo Shaper endodontic rotary file with 300 µm apical diameter and 4% taper (FKG Dentaire, La-Chaux-de-Fonds, Switzerland) was used at 1000 rpm and 1.0 N/cm torque until the root canal filling material was fully removed. Finally, the XP-endo Finisher R endodontic rotary file with 300 µm apical diameter and 0% taper (FKG Dentaire, La-Chaux-de-Fonds, Switzerland) was used at 1000 rpm and 1.0 N/cm torque until the working length as a supplementary cleaning approach to the root canal system, as detailed in the manufacturer’s recommendations. All procedures were performed inside a cabinet with a heater maintained at 37 °C (800-Heater; PlasLabs, Lansing, MI, USA).

Finally, the root canal system was irrigated using a solution consisting of 5 mL of 5.25% NaOCl (Clorox, Oakland, CA, USA), 5 mL of 17% EDTA (SmearClear, SybronEndo, CA, USA), 5 mL of 5.25% NaOCl (Clorox, Oakland, CA, USA) and 5 mL of sterile saline solution (Braun^®^, Melsungen, Germany), with a 0.3-mm endodontic needle inserted 1 mm into the working length. A sonic handpiece driver (EndoActivator^®^, Dentsply Maillefer, Ballaigues, Switzerland) was used to help increase contact between the irrigation solution and the surface of the root canal walls. The non-surgical root canal retreatment was considered finished when the gutta-percha was not visible inside the root canal system under the microscope (OPMI Pico, Zeiss Dental Microscope, Oberkochen, Germany); thereby, the chronometer stopped and the time required to remove the gutta-percha carrier-based root canal filling materials was recorded. Each endodontic rotary file was used in two root canal systems. One clinician performed all of the aforementioned endodontic procedures.

### 2.3. Micro-CT Scanning

All samples underwent a micro-CT scan (Micro-CAT II, Siemens Preclinical Solutions, Knoxville, TN, USA) both prior to and following the non-surgical root canal re-treatment. The following exposure parameters were used: 80 kV, 500 mA, 21 μm isotropic resolution and 360° rotation. An initial micro-CT scan (Micro-CAT II, Siemens Preclinical Solutions, Knoxville, TN, USA) was carried out after root canal treatment with one of the randomly assigned gutta-percha carrier-based root canal filling materials, either GuttaCore (Dentsply Maillefer, Ballaigues, Switzerland) (Figure 1A) or Thermafil (Dentsply Maillefer, Ballaigues, Switzerland) (Figure 1B) (study groups A and B, respectively). A second micro-CT scan (Micro-CAT II, Siemens Preclinical Solutions, Knoxville, TN, USA) was carried out after non-surgical root canal re-treatment with GuttaCore (Dentsply Maillefer, Ballaigues, Switzerland) (Figure 1C) and Thermafil (Dentsply Maillefer, Ballaigues, Switzerland) (Figure 1D). Finally, the remaining root canal filling material was measured and expressed in thirds (Figure 1E,F).

### 2.4. Measurement Procedure

Cobra v.7 was used to automatically reconstruct the micro-CT images (Exxim Computing Corporation, Livermore, CA, USA), and the root canal system’s volume was rendered and the remaining root canal filling material analyzed using Amira 3D software v.6.0 (Thermo Scientific, Agawam, MA, USA). This micro-CT created digital files written in standard tessellation language (STL); this was accomplished using a scattering of points that combined to a tessella network, creating 3D polygonal objects made up of tessellas in the shape of equilateral triangles. These STL digital files were then imported to the FIJI program (National Institutes of Health, Bethesda, MD, USA). The teeth were then aligned by superimposing the STL digital files of each tooth from both before and after the non-surgical root canal re-treatment onto each other. The external surface of the root was used as a reference, and the images were superimposed using the best fit algorithm. Next, the same individual analyzed the volume of the carrier-based root canal filling material within the root canal systems (mm^3^), the volume of any remaining carrier-based root canal filling material (mm^3^), the proportion (%) between the volume of the root canal system and the remaining carrier-based root canal filling material from the coronal, middle and apical thirds of the straight root canal systems and the time taken to perform the non-surgical root canal re-treatment (min).

### 2.5. Statistical Tests

SAS 9.4 was used for the statistical analysis (SAS Institute Inc., Cary, NC, USA). The descriptive analysis included the mean and standard deviation (SD) of quantitative data. For the purposes of comparative statistics the volume of the remaining root canal filling material (mm^3^) and the proportion (%) between the volume of the root canal system and the remaining carrier-based root canal filling material from the coronal, middle and apical thirds of the straight root canal systems were compared with the time (minutes) taken to remove the carrier-based root canal filling material from the coronal, apical and middle thirds of the straight root canal systems. The comparative statistics were calculated using Student’s *t*-test, given that the variables were normally distributed. The cut-off for statistical significance was determined to be *p* < 0.05.

## 3. Results

Table 1 and Figure 2 show the mean and SD values of the pre-operative root canal filling material volume (mm^3^) and remaining post-operative root canal filling material volume (mm^3^) at the coronal third of the straight root canal systems. One NiTi endodontic rotary file (XP-endo Finisher R) was fractured during the non-surgical root canal re-treatment of a tooth in the GuttaCore root canal filling material study group, and this sample was, therefore, withdrawn from the study.

The paired *t*-test found no statistically significant differences (*p* = 0.782) between the volume of remaining root canal filling material in the straight root canal systems in teeth sealed with the GuttaCore root canal filling system (0.12 ± 0.20 mm^3^) and the teeth sealed with the Thermafil root canal filling system (0.11 ± 0.21 mm^3^), at the coronal third of the respective root canal systems (Figure 2).

Table 2 and Figure 3 show the mean and SD values for the pre-operative root canal filling material volume (mm^3^) and remaining post-operative root canal filling material volume (mm^3^) at the middle third of the straight root canal systems.

The paired *t*-test found no statistically significant differences (*p* = 0.838) between the remaining root canal filling material volume of the straight root canal systems in the teeth sealed with the GuttaCore root canal filling system (0.02 ± 0.03 mm^3^) and the teeth sealed with the Thermafil root canal filling system (0.01 ± 0.03 mm^3^), at the middle third of the respective root canal systems (Figure 3).

Table 3 and Figure 4 show the mean and SD values of the pre-operative root canal filling material volume (mm^3^) and the remaining post-operative root canal filling material volume (mm^3^) at the apical third of the straight root canal systems.

The paired *t*-test revealed no statistically significant differences (*p* = 0.882) between the remaining root canal filling material volume of the straight root canal systems in the teeth sealed with the GuttaCore root canal filling system (0.06 ± 0.06 mm^3^) and the teeth sealed with the Thermafil root canal filling system (0.05 ± 0.04 mm^3^), at the apical third of the root canal systems (Figure 4).

Table 4 and Figure 5 show the proportion (%) between the volume of the root canal system and the remaining carrier-based root canal filling material from the coronal, middle and apical thirds of the straight root canal systems.

The paired *t*-test did not show statistically significant differences related to the proportion (%) of the volume of root canal system and remaining GuttaCore root canal filling system and Thermafil root canal filling system at the coronal (*p* = 0.983), middle (*p* = 0.888) and apical thirds (*p* = 0.163) of the straight root canal systems (Figure 5).

Table 5 and Figure 6 show the mean and SD values for the time (min) needed to remove the root canal filling materials from the straight root canal systems.

The paired *t*-test found statistically significant differences (*p* = 0.037) between the time needed to remove the root canal filling material volume from the straight root canal systems of teeth sealed with the GuttaCore root canal filling system (4.72 ± 0.76 min) and the teeth sealed with the Thermafil root canal filling system (5.92 ± 1.42 min) (Figure 6).

## 4. Discussion

The results of this study supported the null hypothesis (H_0_) that there is no difference between the gutta-percha carrier-based root canal filling materials upon removal from straight root canal systems; that being said, statistically significant differences were observed between the amount of time needed to completely remove the gutta-percha carrier-based root canal filling materials from straight root canal systems.

This study analyzed the efficacy of the XP-endo non-surgical root canal re-treatment system in removing GuttaCore and Thermafil gutta-percha carrier-based root canal filling materials with AH Plus sealer from straight root canal systems. GuttaCore is a gutta-percha carrier-based root canal filling material characterized by a rigid central core made up of cross-linked gutta-percha. The heat generated by the oven does not alter the physical properties of the cross-linked gutta-percha in the rigid central core [30], and this material proves to be easier to remove from the root canal system when compared with the Thermafil gutta-percha carrier-based root canal filling material, which has a rigid central core comprised of a plastic or metal carrier [31]. Furthermore, it has been shown that the Thermafil gutta-percha carrier-based root canal filling material requires a longer time to completely remove from the root canal systems when compared with the GuttaCore gutta-percha carrier-based root canal filling material [32,33]. These findings corroborate the results of the present study, in which removal of the GuttaCore gutta-percha carrier-based root canal filling material from straight root canal systems required a statistically significant (*p* = 0.037) shorter time than the Thermafil gutta-percha carrier-based root canal filling material. The longer time needed for the removal of the latter is associated with greater difficulty in removing the plastic carrier of the Thermafil gutta-percha carrier-based root canal filling material [34]. It is, therefore, recommended that the rigid carrier of the Thermafil gutta-percha carrier-based root canal filling material be removed prior to removing the gutta-percha material from the root canal system, resulting in less transportation of the root canal filling material through the apical foramen and lessening the risk of other clinical complications during non-surgical root canal re-treatment using NiTi endodontic rotary instruments [35].

The prognosis of non-surgical root canal re-treatment is dependent on the amount of disinfection achieved on the root canal system [36]; however, remnants of necrotic tissue and/or microorganisms from the primary endodontic infection process may become trapped in residual root canal filling material, leading to potential failure of the non-surgical root canal re-treatment [14,16]. As a result, the clinician must completely remove the root canal filling material to enable the bactericidal properties of the disinfection agents to take effect. Therefore, different non-surgical root canal re-treatment techniques have been proposed; however, Takahashi et al. found no statistically significant differences in the efficacy of non-surgical root canal re-treatment techniques, whether using manual or rotary endodontic instruments, in the removal of root canal filling materials within root canal systems. Nevertheless, some NiTi endodontic rotary systems have been shown to require less time for removal of the root canal filling material within the root canal systems without chloroform when compared with endodontic hand files with chloroform [37]. Therefore, NiTi endodontic rotary instruments pose some advantages with regard to maintenance of root canal anatomy, reduced working time and less clinician fatigue, although they can also lead to extrusion of root canal filling material and debris through the apical foramen [38], alterations in the anatomy of the root canal system and even unexpected fractures of the NiTi endodontic rotary files [34,35]. Fracchia et al. observed unexpected fracture of two NiTi endodontic rotary files during a non-surgical root canal re-treatment with Thermafil root canal filling material at the apical third of the root canal systems; the plastic carrier was also fractured during the non-surgical root canal re-treatment, making the removal procedure more complicated. They also observed that NiTi endodontic rotary instruments used with Thermafil root canal filling material are subjected to higher torsional stress and cyclic fatigue when compared with the GuttaCore Pink root canal filling material, making them more liable to unexpected fracture [39]. This result does not match the findings of the present study, in which one NiTi endodontic rotary file was fractured during the non-surgical endodontic re-treatment with the GuttaCore root canal filling material. Some studies also analyze the effect of gutta-percha solvent agents on the efficacy of removal of root canal filling materials. Willcox and Juhlin reported that success in the non-surgical root canal re-treatment of Thermafil gutta-percha carrier-based root canal filling material primarily depends on the skill of the clinician in removing the carrier above the non-surgical root canal re-treatment technique [38], and Takahashi et al. highlighted that removal of root canal filling materials was faster when a gutta-percha solvent agent was not used [37]. Chemical melting of gutta-percha root canal filling material resulting from the heat generated by non-surgical root canal re-treatment systems produces a thin layer that may adhere to dentinal canal walls [40] or even enter into the isthmus, lateral canals and irregularities [41], making the process of cleaning the root canal more difficult.

Many measurement protocols have been used to analyze the amount of root canal filling material removed from the root canal systems, including tooth splitting [42], dental diaphanization [43], conventional and digital radiography [44] and scanning electron microscopy (SEM) analysis [45]. That being said, tooth splitting is considered an invasive measurement technique that can spread and alter the root canal filling material. Meanwhile, conventional periapical and digital radiography creates 2D images of the 3D root canal system [15]. Currently, micro-CT is the best measurement procedure because it provides 3D-resolution images without destroying the specimen [46]; therefore, micro-CT is the recommended measurement technique for use in endodontic research, especially when the remaining root canal filling material must be analyzed [47,48].

Most studies reported less than 10% of remaining root canal filling material within the root canal systems [6,7]. These results match those obtained in the present study, in both the GuttaCore and Thermafil root canal filling materials’ study. Additionally, some studies have used the XP-endo Finisher NiTi endodontic non-surgical re-treatment instrument as an extra step in root canal re-treatment, showing that the XP-endo Finisher increased the effectiveness of root canal filling material removal [49]. Furthermore, variations in the tip of the XP-endo Finisher R endodontic non-surgical re-treatment instrument make this file more rigid, increasing its effectiveness in removing root canal filling materials that were not able to be removed using conventional non-surgical root canal re-treatment techniques [48,50].

## 5. Conclusions

The XP-endo Finisher non-surgical endodontic re-treatment system removes both GuttaCore and Thermafil gutta-percha carrier-based root canal filling materials from straight root canal systems, with the GuttaCore gutta-percha carrier-based root canal filling material taking less time to be removed.

## Figures and Tables

**Figure 1 jcm-10-01266-f001:**
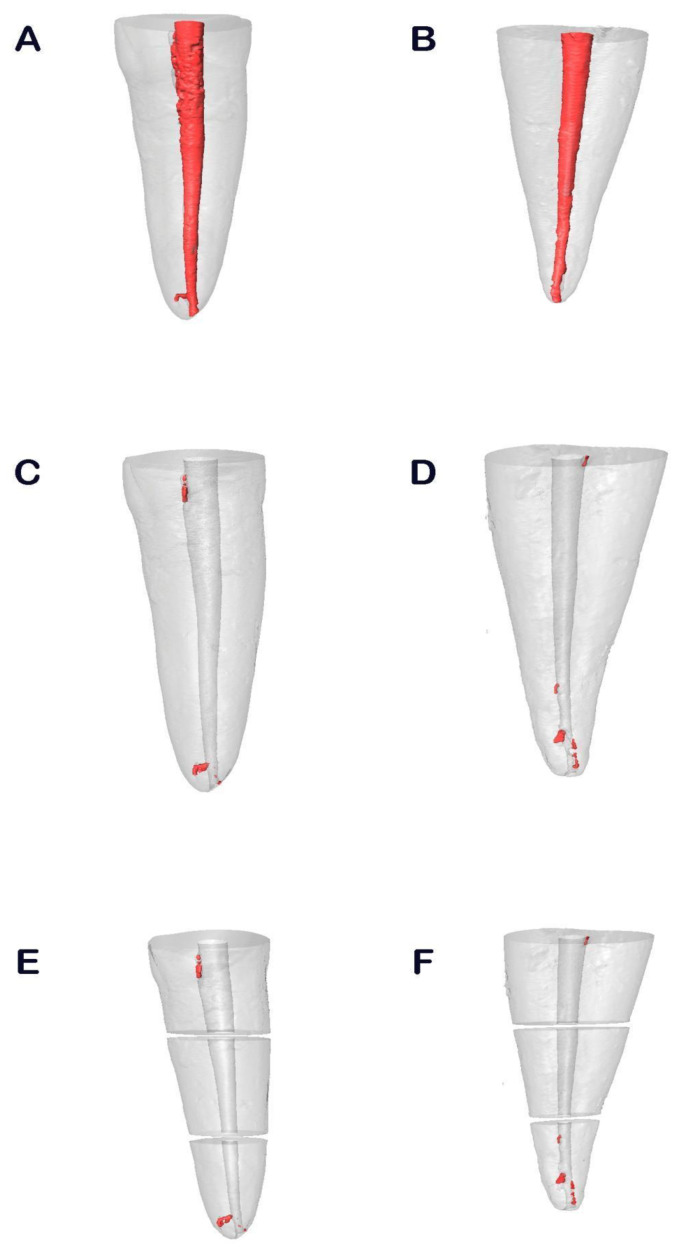
Reconstructed three-dimensional micro-computed tomography (CT) images of the GuttaCore (**A**) and Thermafil root canal filling materials (**B**), after non-surgical root canal re-treatment of GuttaCore (**C**) and Thermafil root canal filling materials (**D**). The results were expressed in thirds both in GuttaCore (**E**) and Thermafil root canal filling materials (**F**).

**Figure 2 jcm-10-01266-f002:**
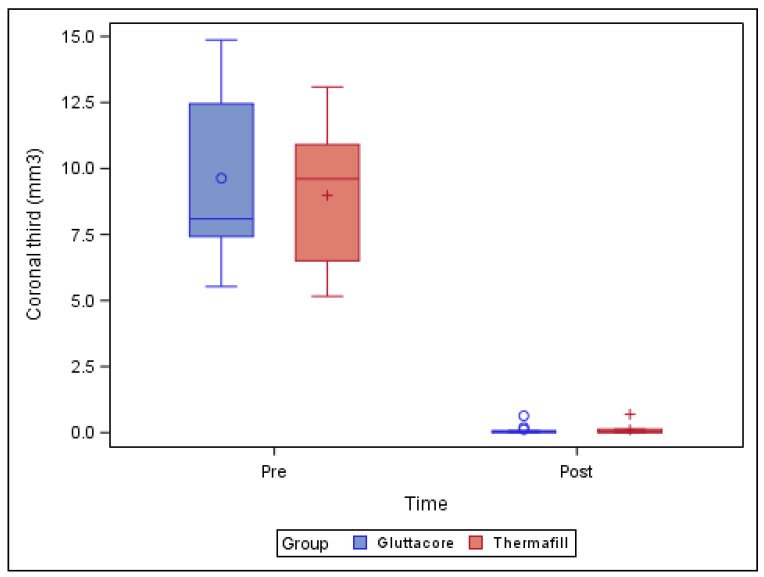
Box plot of the pre-operative root canal filling material volume (mm^3^) and the remaining post-operative root canal filling material volume (mm^3^) at the coronal third of the straight root canal systems, regarding the root canal filling material systems.

**Figure 3 jcm-10-01266-f003:**
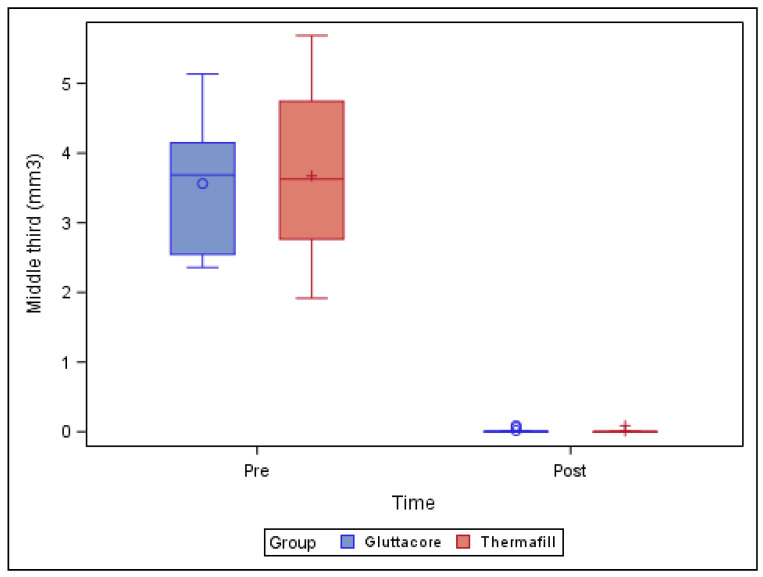
Box plot of the pre-operative root canal filling material volume (mm^3^) and the remaining post-operative root canal filling material volume (mm^3^) at the middle third of the straight root canal systems, categorized by root canal filling material system.

**Figure 4 jcm-10-01266-f004:**
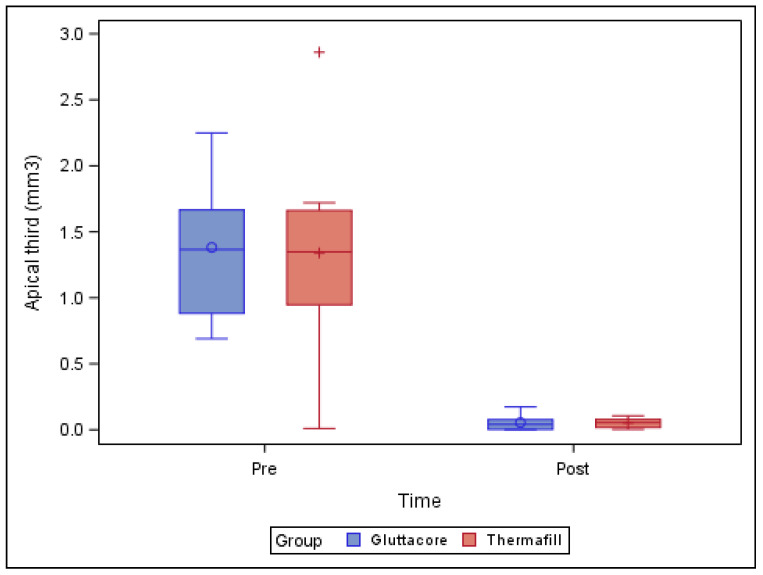
Box plot of the pre-operative root canal filling material volume (mm^3^) and the remaining post-operative root canal filling material volume (mm^3^) at the apical third of the straight root canal systems, categorized by root canal filling material system.

**Figure 5 jcm-10-01266-f005:**
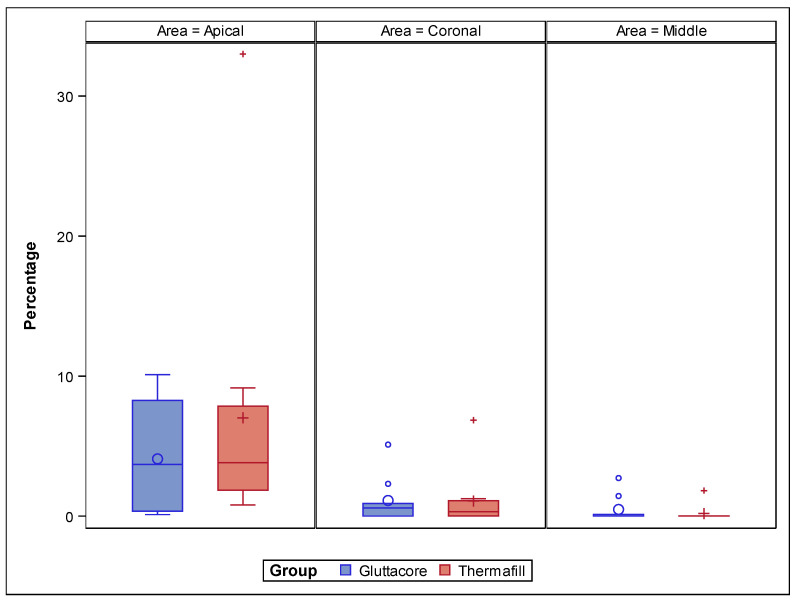
Box plot of the proportion (%) between the volume of the root canal system and the remaining carrier-based root canal filling material from the coronal, middle and apical thirds of the straight root canal systems.

**Figure 6 jcm-10-01266-f006:**
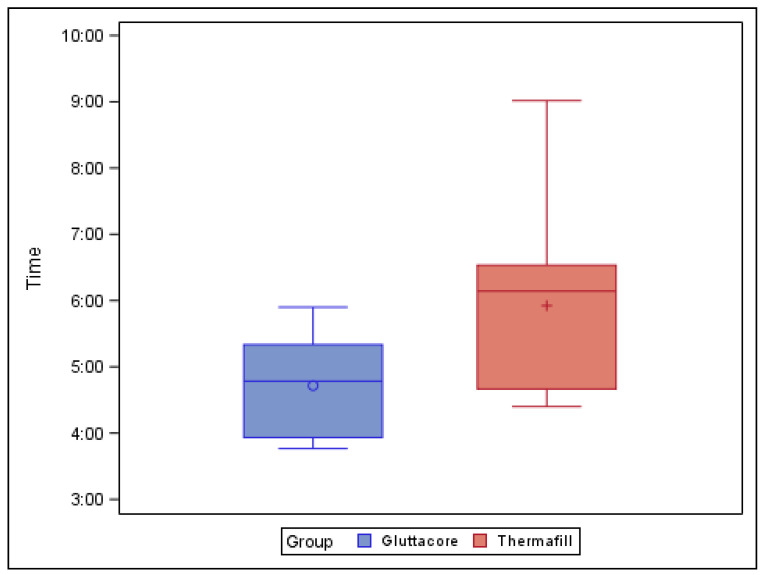
Box plot of the time (min) used to remove root canal filling materials from the straight root canal systems.

**Table 1 jcm-10-01266-t001:** Descriptive analysis of the pre-operative root canal filling material volume (mm^3^) and the remaining post-operative root canal filling material volume (mm^3^) at the coronal third of the root canal systems.

Study Group	Time	*n*	Mean	SD	Minimum	Maximum
GuttaCore	Pre-operative	10	9.63 ^a^	3.16	5.53	14.87
	Post-operative	9	0.12 ^a^	0.20	0.00	0.64
Thermafil	Pre-operative	10	8.98 ^a^	2.55	5.16	13.09
	Post-operative	10	0.11 ^a^	0.21	0.00	00.69

^a^ Statistically significant differences between groups (*p* < 0.05).

**Table 2 jcm-10-01266-t002:** Descriptive analysis of the pre-operative root canal filling material volume (mm^3^) and the remaining post-operative root canal filling material volume (mm^3^) at the middle third of the root canal systems.

Study Group	Time	*n*	Mean	SD	Minimum	Maximum
GuttaCore	Pre-operative	10	3.56 ^a^	0.94	2.36	5.14
	Post-operative	9	0.02 ^a^	0.03	0.00	0.09
Thermafil	Pre-operative	10	3.67 ^a^	1.20	1.92	5.69
	Post-operative	10	0.01 ^a^	0.03	0.00	0.09

^a^ Statistically significant differences between groups (*p* < 0.05).

**Table 3 jcm-10-01266-t003:** Descriptive analysis of the pre-operative root canal filling material volume (mm^3^) and the remaining post-operative root canal filling material volume (mm^3^) at the apical third of the root canal systems.

Study Group	Time	*n*	Mean	SD	Minimum	Maximum
GuttaCore	Pre-operative	10	1.38 ^a^	0.52	0.69	2.25
	Post-operative	9	0.06 ^a^	0.06	0.00	0.17
Thermafil	Pre-operative	10	1.34 ^a^	0.72	0.01	2.86
	Post-operative	10	0.05 ^a^	0.04	0.00	0.11

^a^ Statistically significant differences between groups (*p* < 0.05).

**Table 4 jcm-10-01266-t004:** Descriptive analysis of the proportion (%) between the volume of the root canal system and the remaining carrier-based root canal filling material volume from the coronal, middle and apical thirds.

Study Group	Root Third	Estimate	SD
GuttaCore	Coronal	1.114 ^a^	1.491
	Middle	0.473 ^a^	1.491
	Apical	4.084 ^a^	1.491
Thermafill	Coronal	1.070 ^a^	1.415
	Middle	0.184 ^a^	1.415
	Apical	7.011 ^a^	1.415

^a^ Statistically significant differences between groups (*p* < 0.05).

**Table 5 jcm-10-01266-t005:** Descriptive analysis of the time (min) used to remove the root canal filling materials inside the root canal systems.

Study Group	*n*	Mean	SD	Minimum	Maximum
GuttaCore	9	4.72 ^a^	0.76	3.77	5.90
Thermafill	10	5.92 ^b^	1.42	4.40	9.02

^a,b^ Statistically significant differences between groups (*p* < 0.05).

## Data Availability

Data available on request due to restrictions e.g., privacy or ethical.

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
