# Peer review of "Efficacy of Removing Thermafil and GuttaCore from Straight Root Canal Systems Using a Novel Non-Surgical Root Canal Re-Treatment System: A Micro-Computed Tomography Analysis"

_jcm, 2021, doi:10.3390/jcm10061266_

Round 1

Reviewer 1 Report

In the manuscript entitled “Efficacy of Removing Thermafil and Guttacore From Straight Root Canal Systems Using a Novel Non-Surgical Root Canal Re-Treatment System: A Micro-Computed Tomography Analysis” the authors analysed the efficacy of the XP-endo non-surgical root canal re-treatment system in removing two types of gutta-percha carrier-based root canal filling materials using micro-Ct analysis. I really appreciated the efforts of the authors but I think the manuscript should be rejected. Here you find below some suggestions.

Concerning the time to remove the gutta-percha carrier-based root canal filling materials, this part is not described in materials and methods (for example: did you perform the retreatment using a microscope? when did you decide the end of the treatment to measure the time?). Regarding Micro-CT, did you evaluate the gutta-percha and the sealer? What do you mean for remaining root canal filling material?

In my opinion it is not meaningful to compare the volumes of remaining root canal filling material (despite the use of roots of same lengths but they don’t have the same volume). I think it would be more correct to express the values of remaining canal filling material as a percentage of the filling material. Moreover it would be interesting to compare the coronal, middle and apical parts of the groups with a statistical analysis and then improve the discussion of the article. The results should be completely revised.

Figure 5 is missing.

Author Response

Dear Reviewer 1,

I’m pleased to resubmit the manuscript of the work entitled, “Efficacy of Removing Thermafil and Guttacore From Straight Root Canal Systems Using a Novel Non-Surgical Root Canal Re-Treatment System: A Micro-Computed Tomography Analysis”

Reviewer 1: Concerning the time to remove the gutta-percha carrier-based root canal filling materials, this part is not described in materials and methods (for example: did you perform the retreatment using a microscope? when did you decide the end of the treatment to measure the time?). Regarding Micro-CT, did you evaluate the gutta-percha and the sealer? What do you mean for remaining root canal filling material?

Response: In order to adapt to the reviewer's 1 comments, we have added the time to remove the gutta-percha carrier-based root canal filling materials; moreover, we clarify that the non-surgical root canal retreatment procedures were performed under magnification (OPMI Pico, Zeiss Dental Microscope, Oberkochen, Germany). We also clarify that the non-surgical root canal retreatment was considered finished when the gutta-percha was not visible inside the root canal system under the microscope (OPMI Pico, Zeiss Dental Microscope, Oberkochen, Germany), thereby the chronometer stopped and the time required to remove the gutta-percha carrier-based root canal filling materials was recorded. The measurement procedure based on the comparison of the preoperative and post-operative micro-CT scans provide information related to volume differences, regardless the material (gutta-percha and/or sealer). The authors consider the remaining root canal filling material the gutta-percha and/or sealer remnant in the root canal system after finishing the non-surgical root canal retreatment procedures.

Reviewer 1: In my opinion it is not meaningful to compare the volumes of remaining root canal filling material (despite the use of roots of same lengths but they don’t have the same volume). I think it would be more correct to express the values of remaining canal filling material as a percentage of the filling material. Moreover it would be interesting to compare the coronal, middle and apical parts of the groups with a statistical analysis and then improve the discussion of the article. The results should be completely revised.

Response: In order to adapt to the reviewer's 1 comments, we have included the proportion (%) between the volume of the root canal system and the remaining carrier-based root canal filling material from the coronal, middle and apical thirds of the straight root canal systems. However, no statistical significant differences were shown between the volume of the root canal systems.

We take this opportunity to thank the recommendations and suggestions made by the reviewers to improve the document.

Yours sincerely,

Reviewer 2 Report

Dear Authors,

The article deals with the efficacy in the removal of carrier-based systems from straight root-canals with retreatment systems. The topic is interesting, however I have some doubts regarding the protocol and the clinical relevance of the research.

Abstract

Please check that the names of products are well written.

Keywords

I suggest to modify carrier-based root canal system.

Introduction

This section starts with "At present..", however these references vary from 2003 to 2015. This is not the actual scenario. More recent articles on clinical outcomes have been published and should be cited here.

In the first sentence, secondary treatments are not mentioned as an option after failure of primary treatments. I think it should be added.

Niquel: please correct

XP Endo finisher instruments are not introduced to the readers. I suggest to do it.

M&M

Was the section of the canals evaluated to standardize extracted included samples? I think this is an important variable. 

My major concern is about the use of XP Endo Finisher in this research. You state that it was used "..to shape the root canal system". This instrument was designed for a supplementary cleaning approach and not for shaping the root canal system. Please explain.

"XP -endo non surgical root canal re-treatment system (FKG...)." Which instrument is this? It is unclear to me.

"A sonic device.." Please specify

Which diameter and taper was used to retreat canals? XP_endo finisher r was used till the WL or not? how many times each file was used?

Results

"one niti file was fractured.." which instrument? please specifyPlease remove from the legend of Tab. 1-4 "of the straight root canal systems", References to the manufacturers and "..the remaining ............teeth". This is redundant.

What about statistically significant differences?these are not reported in the Tab. 

Was a comparison between thirds performed? I recommend to add this evaluation and to discuss this aspect. 

Discussion

Please remove mean and SD from this section.

Ref 22: this ref is not correct, the research deals with another topic. I recommend to revise References.

Pag 11: The first sentence is not universally accepted.

"Different non-surgical........method." This is a repetition of the Introduction section.

Ref 30, 31: these research deal with apical extrusion. Have been this evaluated in the present research?

Ref. 39: Sem analysis of the removal of Thermafil system was evaluated by the research of Pirani C et al, Effectiveness of three different...J Endod 2009.

References

Please refer to more recent articles.

I suggest to compare present findings to the research by

Silva E et al. Effectiveness of XP Endo finisher..... Int Endod J 2017.

Mirferendereski M et al. 2009

Pirani C et al IEJ 2017

Pirani C et al Clin Oral Invest 2019

Author Response

Dear Reviewer 2,

I’m pleased to resubmit the manuscript of the work entitled, “Efficacy of Removing Thermafil and Guttacore From Straight Root Canal Systems Using a Novel Non-Surgical Root Canal Re-Treatment System: A Micro-Computed Tomography Analysis”

Reviewer 2: Abstract: Please check that the names of products are well written.

Response: In order to adapt to the reviewer's 2 comments, we have checked the names of the products used.

Reviewer 2: Keywords: I suggest to modify carrier-based root canal system.

Response: In order to adapt to the reviewer's 2 comments, we have changed the keyword.

Reviewer 2: Introduction: This section starts with "At present..", however these references vary from 2003 to 2015. This is not the actual scenario. More recent articles on clinical outcomes have been published and should be cited here.

Response: In order to adapt to the reviewer's 2 comments, we have changed the references.

Reviewer 2: Introduction: In the first sentence, secondary treatments are not mentioned as an option after failure of primary treatments. I think it should be added.

Response: In order to adapt to the reviewer's 2 comments, we have added “non-surgical root canal retreatment”.

Reviewer 2: Introduction: Niquel: please correct

Response: In order to adapt to the reviewer's 2 comments, we have corrected the word.

Reviewer 2: Introduction: XP Endo finisher instruments are not introduced to the readers. I suggest to do it.

Response: In order to adapt to the reviewer's 2 comments, we have introduced the XP-Endo Retreatment System.

Reviewer 2: Material and Methods: Was the section of the canals evaluated to standardize extracted included samples? I think this is an important variable.

Response: In order to adapt to the reviewer's 2 comments, we clarify that the root canal section was analyzed preoperatively by measuring the root canal section in the buccolingual and mesiodistal directions of the digital radiographs resulting slightly oval.

Reviewer 2: Material and Methods: My major concern is about the use of XP Endo Finisher in this research. You state that it was used "..to shape the root canal system". This instrument was designed for a supplementary cleaning approach and not for shaping the root canal system. Please explain.

Response: In order to adapt to the reviewer's 2 comments, we have changed the sentence.

Reviewer 2: Material and Methods: "XP -endo non surgical root canal re-treatment system (FKG...)." Which instrument is this? It is unclear to me.

Response: In order to adapt to the reviewer's 2 comments, we have changed the sentence.

Reviewer 2: Material and Methods: "A sonic device.." Please specify

Response: In order to respond to the reviewer's 2 comments, we have specified the sonic device.

Reviewer 2: Material and Methods: Which diameter and taper was used to retreat canals? XP_endo finisher r was used till the WL or not? how many times each file was used?

Response: In order to adapt to the reviewer's 2 comments, we have clarified the apical diameter and taper of the endodontic rotary files, the working length of the XP-endo finisher R and the times of use of the endodontic rotary files.

Reviewer 2: Results: "one niti file was fractured.." which instrument? please specifyPlease remove from the legend of Tab. 1-4 "of the straight root canal systems", References to the manufacturers and "..the remaining ............teeth". This is redundant.

Response: In order to respond to the reviewer's 2 comments, we have specified the fractured instrument and we have also removed the redundant sentences.

Reviewer 2: Results: What about statistically significant differences?these are not reported in the Tab.

Response: In order to adapt to the reviewer's 2 comments, we have added the statistical significant differences in the Tables.

Reviewer 2: Results: Was a comparison between thirds performed? I recommend to add this evaluation and to discuss this aspect.

Response: In order to adapt to the reviewer's 2 comments, we have done a comparison between thirds, comparing the proportion (%) between the volume of the root canal system and the remaining carrier-based root canal filling material from the coronal, middle and apical thirds of the straight root canal systems. The paired t-test did not show statistically significant differences related to the proportion (%) of the volume of root canal system and remaining GuttaCore root canal filling system and Thermafil root canal filling system at the coronal (p = 0.983), middle (p = 0.888) and apical third (p = 0.163) of the straight root canal systems. In addition,

Reviewer 2: Discusion: Please remove mean and SD from this section.

Response: In order to adapt to the reviewer's 2 comments, we have removed the mean and SD from the Discussion section.

Reviewer 2: Discusion: Ref 22: this ref is not correct, the research deals with another topic. I recommend to revise References.

Response: In order to adapt to the reviewer's 2 comments, we have changed the reference.

Reviewer 2: Discusion: Pag 11: The first sentence is not universally accepted

Response: In order to adapt to the reviewer's 2 comments, we have changed the sentence.

Reviewer 2: Discusion: "Different non-surgical........method." This is a repetition of the Introduction section.

Response: In order to adapt to the reviewer's 2 comments, we have changed the sentence.

Reviewer 2: Discusion: Ref 30, 31: these research deal with apical extrusion. Have been this evaluated in the present research?

Response: In order to adapt to the reviewer's 2 comments, we have revised the references 30 and 31 and the aim of both studies is related to the time required to remove Thermafil and GuttaCore from the root canal systems and the studies are not related to the apical extrusion; therefore, we consider that the references are adequate to the sentence.

Reviewer 2: Discusion: Ref. 39: Sem analysis of the removal of Thermafil system was evaluated by the research of Pirani C et al, Effectiveness of three different...J Endod 2009.

Response: In order to adapt to the reviewer's 2 comments, we have changed the reference.

Reviewer 2: References: Please refer to more recent articles.

Response: In order to adapt to the reviewer's 2 comments, we have included some more recent articles.

Reviewer 2: References: I suggest to compare present findings to the research by

Silva E et al. Effectiveness of XP Endo finisher..... Int Endod J 2017.

Mirferendereski M et al. 2009

Pirani C et al IEJ 2017

Pirani C et al Clin Oral Invest 2019

Response: In order to adapt to the reviewer's 2 comments, we have included the reference “Silva E et al. Effectiveness of XP Endo finisher..... Int Endod J 2017” (Silva EJNL, Belladonna FG, Zuolo AS, Rodrigues E, Ehrhardt IC, Souza EM, De-Deus G. Effectiveness of XP-endo Finisher and XP-endo Finisher R in removing root filling remnants: a micro-CT study. Int Endod J. 2018, 51, 86-91). However, we have also found the reference: Mirfendereski M, Roth K, Fan B, Dubrowski A, Carnahan H, Azarpazhooh A, Basrani B, Torneck CD, Friedman S. Technique acquisition in the use of two thermoplasticized root filling methods by inexperienced dental students: a microcomputed tomography analysis. J Endod. 2009 Nov;35(11):1512-7. doi: 10.1016/j.joen.2009.07.027, but we consider that it is not aligned with the aim of the study. We have not found the reference “Pirani C et al IEJ 2017”. The closest reference that we have found is the following: Pirani C, Friedman S, Gatto MR, Iacono F, Tinarelli V, Gandolfi MG, Prati C. Survival and periapical health after root canal treatment with carrier-based root fillings: five-year retrospective assessment. Int Endod J. 2018 Apr;51 Suppl 3:e178-e188. doi: 10.1111/iej.12757, and we consider that it is not aligned with the aim of the study. Moreover, we have not also found the reference “Pirani C et al Clin Oral Invest 2019”. The closest reference that we have found is the following: Pirani C, Iacono F, Gatto MR, Fitzgibbon RM, Chersoni S, Shemesh H, Prati C. Outcome of secondary root canal treatment filled with Thermafil: a 5-year follow-up of retrospective cohort study. Clin Oral Investig. 2018 Apr;22(3):1363-1373. doi: 10.1007/s00784-017-2229-5, and we also consider that it is not aligned with the aim of the study.

We take this opportunity to thank the recommendations and suggestions made by the reviewers to improve the document.

Yours sincerely,

Round 2

Reviewer 1 Report

I think the manuscript improved and the authors answered my questions.

Best regards

Author Response

Dear Reviewer 1,

I’m pleased to resubmit the manuscript of the work entitled, “Efficacy of Removing Thermafil and Guttacore From Straight Root Canal Systems Using a Novel Non-Surgical Root Canal Re-Treatment System: A Micro-Computed Tomography Analysis”

Reviewer 1: I think the manuscript improved and the authors answered my questions.

Response: We thank the recommendations and suggestions made by the reviewer to improve the document.

Reviewer 2 Report

Dear Authors,

The article has been improved as required, however some major concerns remain. I recommend to revise some points.

  • the references are not always correctly cited in the text. This aspect needs to be revised through all the manuscipt. I.e. ref 32-33 do not refer to apical extrusion, in the text you mention "..lead to apical extrusion of root canal filling....". Which is the ref for this?
  • Ref. 30 remove capital letters
  • ref 38 in incorrect. the article does not deal with radiography. The same for 29, incorrectly associated.
  • I suggest to verify whether all the Ref. in the text are correctly associated.
  • Keywords must be put in alphabetical order
  • in the Introduction section, instruments have been listed, not described as required (DR1, XP-endo Shaper and XP-endo finisher R)
  • captions of the tables need minor revisions.
  • In the introduction section authors mention "Guttapercha carrier based root canal filling materials have become widely used due to easy handling..." I suggest to refer to the suggested articles mentioned in the  the last point of the revision letter. These clinical articles deals with long term outcomes of treatments filled with carrier based material (widely used), and with treatments performed by undergraduates students (easy handling). I think these are relevant.

Author Response

Dear Reviewer 2,

I’m pleased to resubmit the manuscript of the work entitled, “Efficacy of Removing Thermafil and Guttacore From Straight Root Canal Systems Using a Novel Non-Surgical Root Canal Re-Treatment System: A Micro-Computed Tomography Analysis”.

Reviewer 2: the references are not always correctly cited in the text. This aspect needs to be revised through all the manuscipt. I.e. ref 32-33 do not refer to apical extrusion, in the text you mention "..lead to apical extrusion of root canal filling....". Which is the ref for this?

Response: In order to adapt to the reviewer's 2 comments, we have included a reference.

Reviewer 2: Ref. 30 remove capital letters

Response: In order to adapt to the reviewer's 2 comments, we have removed capital letters.

Reviewer 2: ref 38 in incorrect. the article does not deal with radiography.

Response: In order to adapt to the reviewer's 2 comments, we have changed the reference.

Reviewer 2: The same for 29, incorrectly associated.

Response: In order to adapt to the reviewer's 2 comments, we have added a reference and changed reference 29.

Reviewer 2: I suggest to verify whether all the Ref. in the text are correctly associated.

Response: In order to adapt to the reviewer's 2 comments, we have revised the references in the text.

Reviewer 2: Keywords must be put in alphabetical order

Response: In order to adapt to the reviewer's 2 comments, we have order the keywords in alphabetical order.

Reviewer 2: in the Introduction section, instruments have been listed, not described as required (DR1, XP-endo Shaper and XP-endo finisher R)

Response: In order to adapt to the reviewer's 2 comments, we have described the XP-Endo-Retreatment System.

Reviewer 2: captions of the tables need minor revisions.

Response: In order to adapt to the reviewer's 2 comments, we have revised the captions of the Tables.

Reviewer 2: In the introduction section authors mention "Guttapercha carrier based root canal filling materials have become widely used due to easy handling..." I suggest to refer to the suggested articles mentioned in the  the last point of the revision letter. These clinical articles deals with long term outcomes of treatments filled with carrier based material (widely used), and with treatments performed by undergraduates students (easy handling). I think these are relevant.

Response: In order to adapt to the reviewer's 2 comments, we have included the mentioned articles in the Introduction section.

We take this opportunity to thank the recommendations and suggestions made by the reviewers to improve the document.

Yours sincerely,

This manuscript is a resubmission of an earlier submission. The following is a list of the peer review reports and author responses from that submission.